# *Helicobacter pylori* inhibition by antimicrobial photodynamic therapy of methylene blue and porphyrin

Daotan Zhao,[1] Yina Ba,[2] Ping Xu,[1] Shuyang Sun[1]

**ABSTRACT** *Helicobacter pylori* is the pathogen responsible for peptic ulcers and gastritis, affecting half of the population around the world. The infection of *H. pylori* is commonly treated with a combination of antibiotics, but the effectiveness is decreasing due to the rising multidrug resistance in recent years. Antimicrobial photodynamic therapy is a promising alternative for *H. pylori* eradication, where light irradiation activates photosensitizer to produce reactive oxygen species *in situ*, thus inducing cell death through irreversible oxidative damage. Methylene blue is widely used as a coloring agent for chromoendoscopy and also proved to be an effective photosensitizer against bacteria, fungi, viruses, and cancer cells. Additionally, photosensitizer porphyrin is naturally produced and accumulated within *H. pylori* cells. Here, we investigated the inhibitory effects of photosensitizer methylene blue and porphyrin against *H. pylori* free-swimming cells and sessile biofilms. Additionally, we used suckling pig as an animal model to determine the efficiency of antimicrobial photodynamic therapy in the treatment of *H. pylori* infection. Results showed that free-swimming *H. pylori* was eliminated effectively, either by 10 µM of methylene blue under the light irradiation of 660 nm or by endogenous porphyrin activated under the light irradiation of 410 nm. In contrast, *H. pylori* biofilms exhibited resistance. Moreover, *in vivo* experiments showed that antimicrobial photodynamic therapy could inhibit *H. pylori* in the stomach of sucking pigs, but occasional recurrent infections were present.

**IMPORTANCE** The rising prevalence of multidrug-resistant *Helicobacter pylori* severely compromises conventional antibiotic therapies, necessitating novel treatment strategies. Antimicrobial photodynamic therapy (aPDT) offers a promising alternative by using light-activated reactive oxygen species to kill bacteria. Here, we show that both exogenous methylene blue and endogenous porphyrin can effectively eliminate free-swimming *H. pylori* under light irradiation. Using a physiologically relevant suckling pig model, we demonstrate that aPDT reduces gastric *H. pylori in vivo*, although recurrent infection was observed. This study provides critical preclinical evidence for aPDT as a potential adjunctive strategy in the treatment of *H. pylori* infection, while highlighting the key challenges of biofilm resistance.

**KEYWORDS** *Helicobacter pylori*, antimicrobial photodynamic therapy, methylene blue, porphyrin

**Peer Reviewer** Francisco Avilés, Instituto Mexicano del Seguro Social, México, Mexico

Address correspondence to Shuyang Sun, sun_shuyang@sjtu.edu.cn.

Daotan Zhao and Yina Ba contributed equally to this article. The author order was determined based on primary leadership of the project and extent of contributions, with Daotan Zhao listed first for leading the experimental work and manuscript preparation.

The authors declare no conflict of interest.

*H*elicobacter pylori was first identified in the 1980s in the stomachs of patients with gastritis or peptic ulcers (1, 2). Research advances have further characterized *H. pylori* as a microaerobic gram-negative pathogen that could colonize the gastric mucus layer or adhere to gastric epithelial cells, which weakens the mucosa lining of stomach, and thus the penetrated acid irritates underlying tissue and causes an ulcer. *H. pylori* infection usually occurs in childhood and develops into chronic progressive gastritis throughout life. Approximately 1%–10% of infected individuals develop clinical

symptoms, including peptic ulcer disease, gastric atrophy, gastric mucosal enteroplasia, and eventually gastric cancer or mucosa-associated lymphoid tissue lymphoma (3, 4). To date, *H. pylori* has infected approximately 4.4 billion people worldwide, and the World Health Organization has ranked it as one of the top three threats to public health in the 21st century (5, 6).

The high infection rate and misuse of antibiotics lead to increased antibiotic resistance of *H. pylori*. It has been reported that the efficacy of standard triple therapy decreases significantly to only about 70%, and the recurrence rate increases gradually (7). Additionally, *H. pylori* has been observed to form biofilms in gastric mucosa and aggregates in gastric glands, which exhibit higher resistance to antibiotics and contribute to the recurrent infection (8, 9). The persistence of *H. pylori* leads to a demand for novel clinical treatment options.

Antimicrobial photodynamic therapy (aPDT) is a hotspot in the field of clinical antimicrobial applications over the past decade, where a clinically safe photosensitizer is activated by light irradiation to generate reactive oxygen species (ROS) *in situ*, thereby inducing cell death through irreversible oxidative damage. Compared to conventional therapy of antibiotics, aPDT takes the advantages of rapid efficacy, high selectivity, and low toxicity (10).

Methylene blue (MB) is a water-soluble pigment and widely used in chromoendoscopy. Clinical guidelines recommend topical application of MB for pigmented endoscopy to detect gastrointestinal dysplastic lesions (11). In addition, the FDA has approved MB for the treatment of methemoglobinemia via intravenous administration (12). Therefore, MB has been accepted as a safe coloring agent in clinical applications (13). Meanwhile, MB has been identified as a photodynamic antimicrobial agent, with an absorption spectrum peaked at 660 nm (14). For example, Cabral et al. (15) used MB-mediated aPDT to successfully treat canine dermatophytosis. Ragàs et al. (16) demonstrated effective inhibition of *Acinetobacter baumannii* by MB-mediated aPDT in burned mice model. Baeshen et al. (17) showed that MB-mediated aPDT had an inhibitory effect against periodontal microorganisms in gingivitis, such as *Porphyromonas gingivalis* and *Tannatobacterium contortum*. It has been shown that typical pathogenic bacteria, such as *Escherichia coli* and *Bacillus cereus*, can be effectively killed *in vitro* by light-activated MB at the concentration less than 1 mM (18).

Antimicrobial blue light (aBL) within the spectral range of 400–470 nm exerts intrinsic antimicrobial activity by activating endogenous photosensitizing chromophores (e.g., porphyrins) in diverse microorganisms (19). This process involves the generation of ROS upon blue light excitation, leading to multi-target effects such as lipid peroxidation of bacterial membranes, protein denaturation, and DNA damage (20). Porphyrin is a class of biological pigment molecules. Currently, coproporphyrins, protoporphyrins, and uroporphyrins have been found in microorganisms, capable of absorbing light and producing ROS (21). Murdoch et al. (22) showed a significant reduction of yeast colony-forming units by 405 nm irradiation. Ashkenazi et al. (23) demonstrated that endogenous coproporphyrins play a major role in the photoinactivation of *Propionibacterium acnes* by irradiation with blue light (407–420 nm). *H. pylori* accumulates porphyrins during its growth, including protoporphyrin IX and coproporphyrins I and III, and the porphyrin mixture of *H. pylori* shows a strong absorption around 405 nm (24, 25).

Here, MB-mediated aPDT and aBL were applied to treat *H. pylori in vitro*, and the inhibitory effects were evaluated by models of free-swimming cells and sessile biofilms. Furthermore, suckling pig was used as an animal model to determine the efficiency of aPDT in the treatment of *H. pylori in vivo*. The results show that aPDT and aBL can effectively eradicate *H. pylori* in suspensions, while the effects were more limited for biofilms. *H. pylori* was inhibited by aPDT in the stomachs of infected animals, but not eliminated, as recurrent infections were detected.

## MATERIALS AND METHODS

### *H. pylori* and culture conditions

*H. pylori* ATCC 43504 was cultured using Columbia blood agar plates at 37°C under micro-oxygenated conditions (AnaeroPack, MGC) for 3 days before aPDT treatment. For biofilm development, *H. pylori* was resuspended in brain heart infusion broth containing 5% fetal bovine serum at the concentration of $10^7$ cells/mL, and 0.4 mL of *H. pylori* suspension was added to a well in 24-well microtiter plate and incubated at 37°C under micro-oxygenated conditions for 3 days (8).

### Photodynamic treatment of *H. pylori* suspension *in vitro*

*H. pylori* was harvested from Columbia blood agar and resuspended to $10^8$ cells/mL using phosphate-buffered saline (PBS), and 0.4 mL of the suspension was added to a well in 24-well microtiter plate, where the depth of the bacterial suspension was standardized to 2 mm to ensure consistent light irradiation and to approximate the optical path length relevant to the gastric environment, and then the *H. pylori* suspension was treated using 10 µM, 100 µM, and 1 mM MB and irradiated with 660 nm LED lamp, 40 mW/cm² for 4 min (light dose 9.6 J/cm², the same as below), or directly irradiated with 410 nm LED lamp, 9.6 J/cm². The antibacterial effect was quantitatively measured by CFU counts on Columbia blood agar plates and flow cytometry counts after staining with SYTO 9 and propidium iodide.

### Photodynamic treatment of *H. pylori* biofilms *in vitro*

The *H. pylori* biofilm was prepared at 37°C under micro-oxygenation for 3 days, after gently removing the medium and adding 0.4 ml of PBS, and biofilm formed on the bottom of microtiter plate well was treated by 10 µM, 100 µM, and 1 mM MB and irradiated with 660 nm LED lamp, 9.6 J/cm², or directly irradiated with 410 nm-LED lamp, 9.6 J/cm².

The biomass of *H. pylori* biofilms was evaluated using the MTT assay, where 3-(4,5-dimethylthiazol-2-yl)-2,5-diphenyltetrazolium bromide (MTT, CAS: 298-93-1, J&K SCIENTIFIC) was converted to insoluble purple formazan crystals by intracellular dehydrogenases of viable bacteria. After light treatment, MTT reagent was added to *H. pylori* biofilms at a final concentration of 0.5 mg/mL and incubated for 4 h at 37°C under micro-oxygenated conditions. Subsequently, the supernatant was carefully removed, and the formazan crystals were then dissolved by adding 1 mL DMSO and measured at 490 nm using microplate reader.

The antibacterial effect was also measured by laser scanning confocal microscopy (IXplore SpinSR, Olympus) after staining with SYTO 9 and propidium iodide. The microimages were further analyzed by Imaris (Version 10.2, Oxford Instruments).

### Photodynamic treatment of *H. pylori*-infected suckling pigs

Given the anatomical and functional homology in the digestive system between *Sus scrofa domesticus* and *Homo sapiens*, combined with parallel dietary plasticity, the suckling pig model was used to test aPDT of *H. pylori in vivo*. Five 3-week-old suckling pigs were administered orally with 16 mg/kg of indomethacin once a day for 3 consecutive days to facilitate *H. pylori* colonization. For *H. pylori* inoculation, each suckling pig fasted for 4 h and then was fed with 2 mL of 0.6 M NaHCO₃ solution, followed by 2 mL of $10^9$ CFU/mL *H. pylori* in PBS to infect, by squirting the suspension into their open mouths and then allowing the normal swallowing reflex, repeated once a day for 3 consecutive days (26). Gastroscopy was operated after fasting for 12 h. Each suckling pig was anesthetized with an intramuscular injection of 5 mg/kg Zoletil (Virbac), and then pronase (20,000 U, Tidepharm) was dissolved in 50 mL PBS and administered through gastroscopy system. After 15 min, 50 mL of 15 µM MB solution was sprayed into the stomach through the gastroscopy system. Gastric lesser curvature, gastric body, and

pylorus were exposed to 660 nm LED lamp at approximately 40 mW/cm$^2$ for 10 min (light dose 24 J/cm$^2$), using a flexible optical fiber passed through the biopsy channel of the endoscope. After aPDT treatment, the biopsies of gastric lesser curvature, gastric body, and pylorus were taken from the light-exposed areas (27) and further tested for *H. pylori* presence using the Urease Detection Kit (Anxin Biotech) as per manufacturer's instructions (28). In addition, samples of within-subject pre-treatment control were collected from standardized anatomical sites before aPDT.

## RESULTS

### Eradication of *H. pylori* in suspension by aPDT and aBL

To test the inhibitory effect of MB and porphyrin on *H. pylori*, light-treated resuspended cells were serially diluted and plated on Columbia blood agar. In the unilluminated control, *H. pylori* CFU declined when MB concentration increased, and less than 1% of *H. pylori* was observed at 1 mM MB ($P < 0.001$) (Fig. 1A). Light irradiation at 660 nm had no effect on *H. pylori* without MB, whereas the samples to which 10 µM MB was added were free of *H. pylori* growth immediately after the light treatment ($P < 0.001$) (Fig. 1A). Since the porphyrins produced by *H. pylori* were excited by 410 nm LED lamp to produce singlet oxygen, there was no *H. pylori* colony formed after 410 nm LED lamp treatment ($P < 0.001$) (Fig. 1B).

To further investigate the antiseptic effects of MB and porphyrin, the light-treated *H. pylori* was stained with SYTO 9 and propidium iodide and subsequently counted by flow cytometry. The *H. pylori* suspension contained approximately one-third propidium

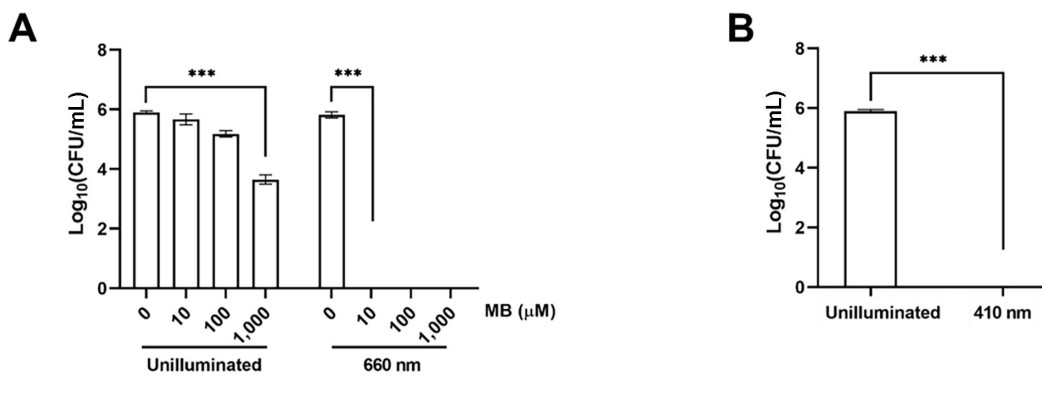

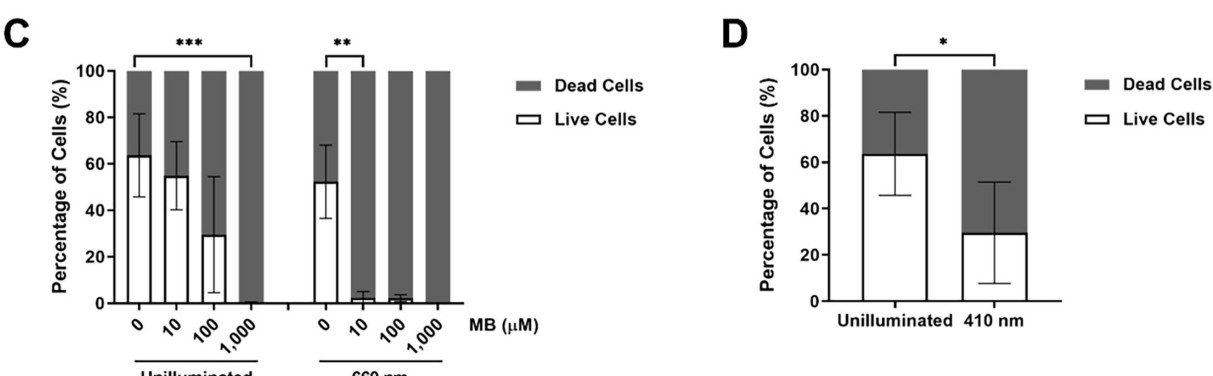

**FIG 1** Inhibitory effects of MB and porphyrin on *H. pylori* free-swimming cells *in vitro*. (A) CFU of *H. pylori* treated with MB under 660 nm irradiation, (B) CFU of *H. pylori* when endogenous porphyrin was activated by 410 nm light, (C) live and dead cell counts of *H. pylori* treated with MB under 660 nm irradiation, and (D) live and dead cell counts of *H. pylori* when endogenous porphyrin was activated by 410 nm light. The SYTO 9 and propidium iodide were used to stain live and dead cells of *H. pylori* in flow cytometry. Experiments were repeated three times, statistics were analyzed using one-way analysis of variance (A and C) and Student's *t*-test (B and D), *$P < 0.05$, **$P < 0.01$, ***$P < 0.001$.

iodide-stained cells, indicating that the staining process damaged the integrity of *H. pylori* membrane. In the unilluminated control, the percentage of viable *H. pylori* decreased with increased MB concentration, to less than 1% at 1 mM MB ($P < 0.001$) (Fig. 1C). When exposed to 660 nm LED lamp irradiation, 10 μM MB almost eliminated viable *H. pylori* ($P < 0.01$) (Fig. 1C). In comparison, porphyrins activated by 410 nm illumination halved the percentage of viable cells ($P < 0.05$) (Fig. 1D).

Both CFU and flow cytometry counting revealed the inhibitory effects of MB and porphyrin under light irradiation, but discrepancies were observed between measuring methods, as there were some viable *H. pylori* cells detected by flow cytometry while no colonies formed on plates. It is probable that DNA damage caused by ROS prevented cells from growing to form colonies, but the integrity of the cell membrane was not disrupted to enable propidium iodide staining, which was particularly relevant to endogenous porphyrins.

## Inhibition of *H. pylori* biofilms by aPDT and aBL

To determine the efficiency of aPDT and aBL against *H. pylori* biofilms, *H. pylori* biofilms were cultured *in vitro* and subsequently treated with light-activated exogenous MB or endogenous porphyrins. The *H. pylori* cells in biofilm were aggregated and could not form colonies by individual cells, and the biofilm biomass was quantified by MTT assay. Without or with MB, there was essentially no difference between the unilluminated control and the 660 nm LED lamp groups of the *H. pylori* biofilms (Fig. 2A). When the *H. pylori* biofilm was treated by 410 nm LED lamp, singlet oxygen produced by irritated porphyrin reduced MTT readings of biofilm by 50% (Fig. 2B), but the difference between the unilluminated control and the 410 nm LED lamp group was not statistically significant.

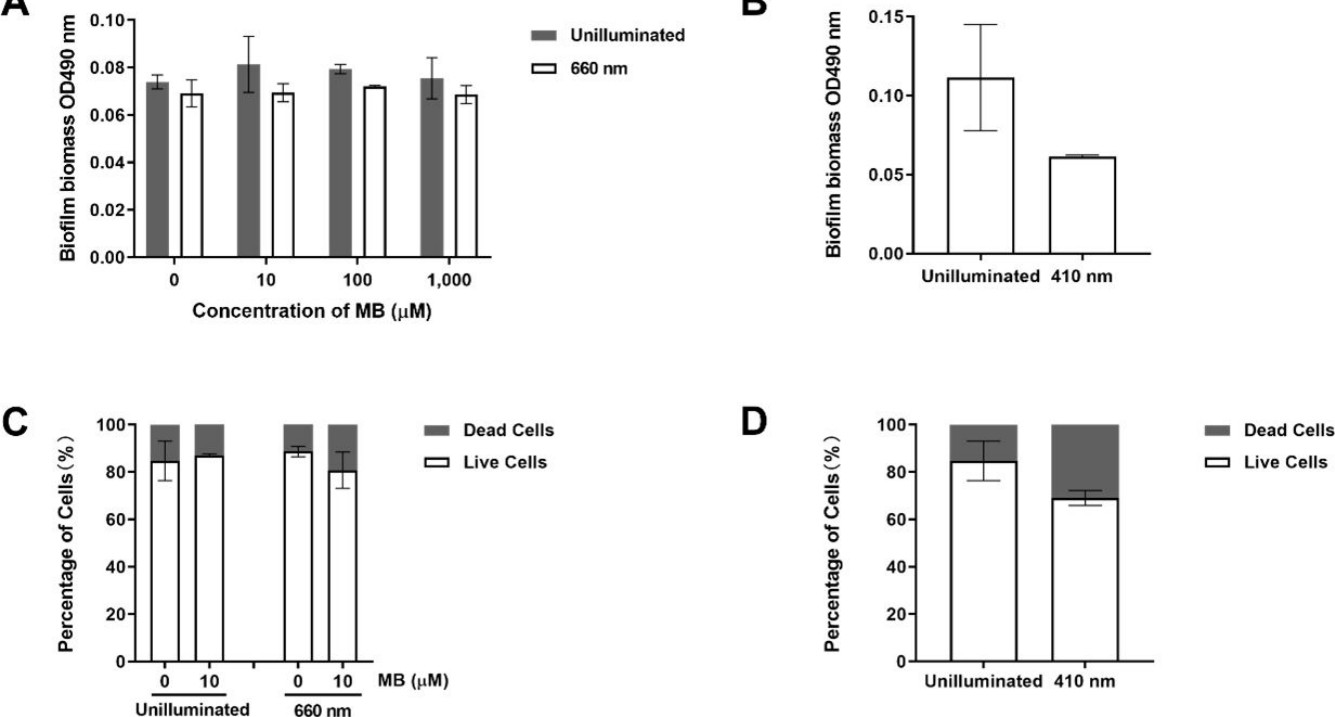

**FIG 2** Inhibitory effects of MB and porphyrin on *H. pylori* biofilms *in vitro*. (A) *H. pylori* biofilms treated with MB and 660 nm LED lamp and quantified by MTT assay, (B) *H. pylori* biofilms inhibited by porphyrins with 410 nm LED lamp and quantified by MTT assay, (C) live and dead cell counts of *H. pylori* biofilms treated with MB and 660 nm LED lamp, and (D) live and dead cell counts of *H. pylori* biofilms treated with 410 nm LED lamp. Experiments were repeated three times, statistics were analyzed using two-way analysis of variance (A and C) and Student's *t*-test (B and D).

Furthermore, *H. pylori* biofilms were observed by laser scanning confocal microscopy. The SYTO 9-stained *H. pylori* was predominant in the untreated control (Fig. 3A), and a similar pattern was shown in the treatment of 10 µM MB (Fig. 3B), the treatment of the 660 nm LED lamp (Fig. 3C), and the treatment of 10 µM MB plus 660 nm LED lamp (Fig. 3D), indicating that *H. pylori* in biofilms was not sensitive to the MB-based aPDT. The percentages of live and dead cell counts analyzed by Imaris also showed no significant differences between control and MB-treated *H. pylori* biofilms, with or without 660 nm irradiation (Fig. 2C). The *H. pylori* biofilms treated with 100 µM and 1 mM MB under 660 nm irradiation were also similar to the untreated control (data not shown), consistent with the MTT assay.

When *H. pylori* biofilms were exposed to the 410 nm LED lamp, microimages showed an increase of the propidium iodide-stained dead cells (Fig. 3E), which was further confirmed by cell counts using Imaris, where dead cells increased from 20% in the unilluminated control to more than 30% under 410 nm irradiation, but the difference was not statistically significant (Fig. 2D). From the side views of the 410 nm LED lamp treated samples, it can be observed that the top layer of *H. pylori* biofilms contains more red fluorescent signals, while the bottom layer is dominated by green fluorescent signals, indicating that aBL took effects on the surface layer of biofilms and *H. pylori* cells inside of biofilms were protected by light blocking (Fig. 3F).

## Treat *H. pylori* infection by aPDT *in vivo*

The suckling pig animal model was used to investigate the effects of aPDT *in vivo*, and *H. pylori* was determined by rapid urease assay. Because 410 nm light could not penetrate the mucus of stomach efficiently (data not shown), only the combination of MB and 660 nm light was applied for *in vivo* test. After *H. pylori* inoculation, the pathological tissue sections showed that *H. pylori* colonized the stomach pylorus (Fig. 4A), and all pigs were positive in rapid urease test (Table 1). To identify the short-term effects of aPDT, samples were collected immediately from the pylorus of two pigs (S1 and S2). *H. pylori* was not detected as no color development from urease assay was observed (Table 1) and no brown-stained *H. pylori* cells were observed in tissue section (Fig. 4B). For long-term effects of aPDT, samples were collected from gastric body, lesser curvature, and pylorus of three pigs (L1, L2, and L3) 1 week after treatment, and weak urease color development was observed in most samples, indicating that *H. pylori* was inhibited but not eliminated, and there were recurrent infections (Table 1).

## DISCUSSION

Due to the increased antibiotic resistance and high recurrence rate of *H. pylori*, the traditional antibiotic therapy is facing severe challenges, and aPDT is a novel solution rising in recent years. This study shows that 10 µM MB activated by 660 nm irradiation can effectively eliminate *H. pylori* in bacterial suspension through oxidative damage. Since the MB solution used for chromoendoscopy is typically 10–20 mM, the *H. pylori* inhibition by low concentration of MB makes aPDT treatment *in vivo* possible. Meanwhile, the endogenous photosensitizer of porphyrin can also inhibit *H. pylori* cells under 410 nm illumination. These observations are consistent with previous studies (24, 27, 29–31). However, when *H. pylori* biofilms were exposed to aPDT and aBL, 660 nm light-activated MB could not kill *H. pylori* effectively, while 410 nm LED lamp may only damage *H. pylori* on the surface of biofilms. It is possible that MB could not penetrate the biofilm matrix and approach *H. pylori* cells efficiently during the limited time of aPDT, and 410 nm light activates porphyrin within surface cells but could not pass through the depth of *H. pylori* biofilms. Our study observed that *H. pylori* within biofilms exhibited markedly higher resistance to photodynamic inactivation compared to their planktonic counterparts. This reduced efficacy is likely multifactorial: bacteria within biofilms often enter a metabolically dormant state, which may downregulate the heme biosynthetic pathway and result in lower accumulation of endogenous photosensitizers. Furthermore, the extracellular polymeric substance matrix acts as a physical barrier, scattering incident

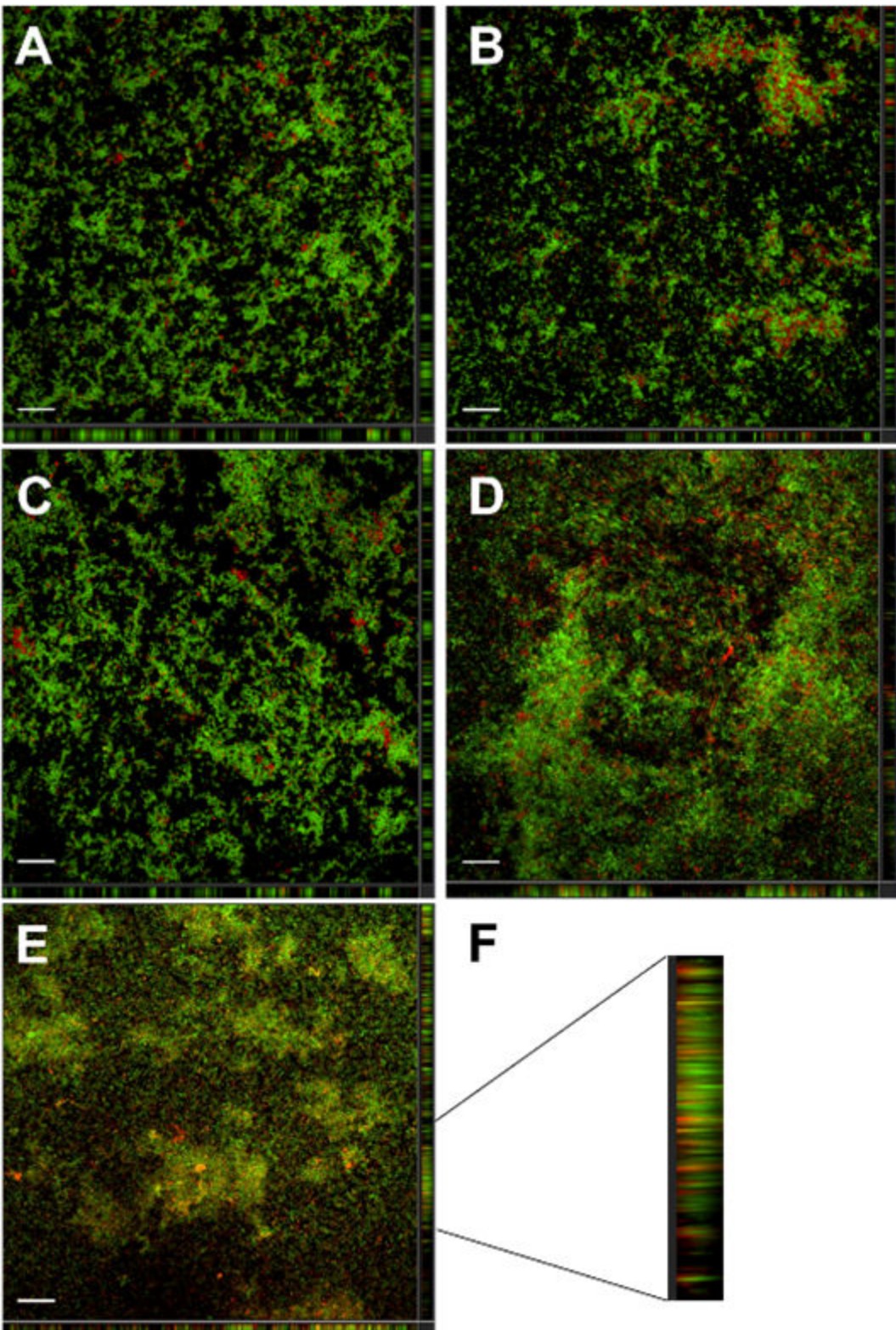

**FIG 3** Microscopic images of the inhibitory effects of MB and porphyrin on *H. pylori* biofilms. The z-stacked microimages are presented with xz and yz side views, and the scale bar is 50 µm. The green pixels represent live cells, and the red pixels represent dead cells. (A) *H. pylori* biofilms untreated, (B) *H. pylori* biofilms treated with 10 µM MB, (C) *H. pylori* biofilms treated with 660 nm LED lamp, (D) *H. pylori* biofilms treated with 10 µM MB and 660 nm LED lamp, (E) *H. pylori* biofilms treated with 410 nm LED lamp, and (F) the yz side views of *H. pylori* biofilms treated with 410 nm LED lamp. Experiments were repeated three times.

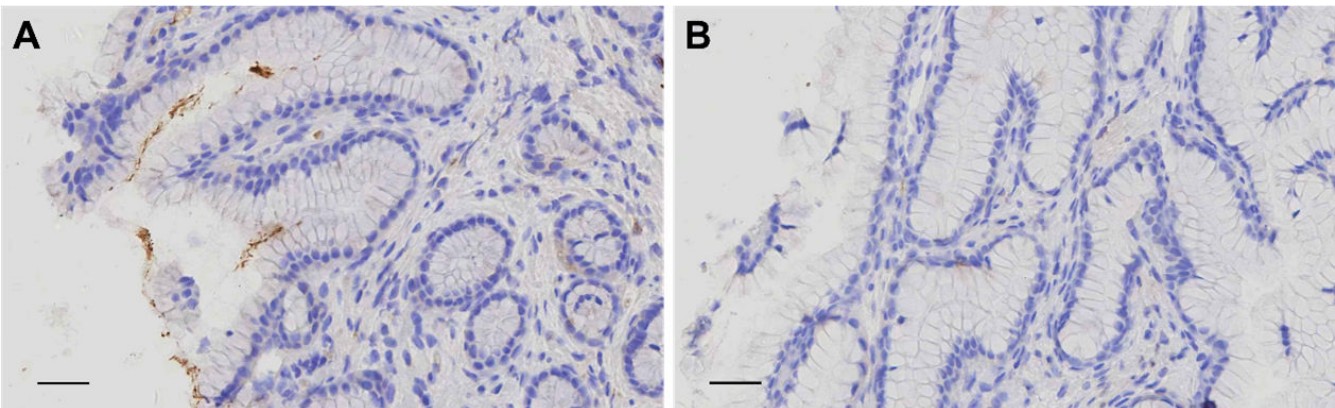

**FIG 4** The pathological tissue section of *H. pylori*-infected pig stomach pylorus. *H. pylori* was brown by immunohistochemistry staining. The scale bar is 20 µm. (A) *H. pylori* colonized the pylorus. (B) *H. pylori* undetected in pylorus mucosa after aPDT treatment exposed to 660 nm LED lamp, 24 J/cm².

light and impeding the penetration of reactive oxygen species (10). Due to the protective characters of biofilms, strategies that induce biofilm decomposition and diffusion may be introduced as a pre-treatment step to improve the efficiency of aPDT. For instance, it has been reported that NO (32) and c-di-GMP inhibitors (33) could induce biofilm decomposition and enhance the bactericidal effects of antibiotics.

In addition, we conducted aPDT *in vivo*, where MB and 660 nm irradiation were applied to treat suckling pigs infected with *H. pylori*. The results show that aPDT could reduce *H. pylori* to undetectable levels, but not eliminated, and recurrent infections of *H. pylori* might occur. The persistence of *H. pylori* infection may be attributed to the biofilms associated with stomach tissues. Previous studies have demonstrated that *H. pylori* can indeed establish biofilms *in vivo*, which promotes sustained bacterial colonization within the gastric mucosa and reduces the effectiveness of antimicrobial treatments against this pathogen (8, 9, 34, 35). Moreover, the efficiency of aPDT depends on illumination, which may be limited by the structure of the gastroscopy system and irregular shape of the stomach, leaving shelter niches for *H. pylori* and causing recurrent infections. It has been proposed that the endoscopic capsule controlled by magnets may take the advantages of size and movement, approach the gastric lesions precisely, and deliver better illumination for aPDT (36). The simplified operation of capsule endoscopy also enables repeated aPDT to eradicate *H. pylori* and prevent recurrent infections. It is also important to acknowledge that the capacity for endogenous porphyrin accumulation is not exclusive to pathogenic *H. pylori*. The heme biosynthetic pathway is evolutionarily conserved across a diverse array of microorganisms, including

**TABLE 1** Determine the aPDT effects against *H. pylori* infection by rapid urease test[a]

| Pig no. | Sample site | Before treatment | After treatment |
| --- | --- | --- | --- |
| S1 | Pylorus | ++ | − |
| S2 | Pylorus | ++ | − |
| L1 | Gastric lesser curvature | ++ | ++ |
|  | Pylorus | ++ | − |
|  | Gastric body | ++ | + |
| L2 | Gastric lesser curvature | ++ | + |
|  | Pylorus | ++ | + |
|  | Gastric body | ++ | − |
| L3 | Gastric lesser curvature | ++ | + |
|  | Pylorus | ++ | − |
|  | Gastric body | ++ | + |

[a]The urease produced by *H. pylori* leads to development of red color: (−) yellow, no color change, (+) orange color, (++) red color.

many commensal members of the gastrointestinal microbiota (37). Consequently, aPDT utilizing broad-spectrum visible light may inherently lack absolute species-specificity, potentially leading to "bystander effects" on the healthy gastric microbiome. Although *H. pylori* has been reported to accumulate significantly higher levels of photoactive porphyrins compared to other gut bacteria under specific conditions (25), future clinical protocols must carefully evaluate the ecological impact of aPDT on the host microbial balance to minimize dysbiosis.

Additionally, we acknowledged the constraints of the limited animal sample size, which was insufficient to yield statistical results. Large-scale trials are necessary to verify the therapeutic effect of aPDT-antibiotic combination therapies in further exploration. While the present study has established the feasibility of aPDT for gastric applications, the observed results revealed that aPDT alone may not be sufficient for complete *H. pylori* eradication within the complex gastric environment. A combination of aPDT and conventional antibiotic regimens may be a probable direction for future investigation, which is expected to enhance antibacterial effect or shorten the dosing time of antibiotic (38).

## Conclusion

The results of this study show that aPDT by 660 nm LED lamp activated MB or aBL by 410 nm LED lamp activated porphyrin can effectively photo-inactivate *H. pylori* in suspensions, whereas efficacies of both treatments are more limited for *H. pylori* biofilms. Additionally, aPDT reduces *H. pylori* effectively *in vivo*, but some persistent cells may cause recurrent infections. Therefore, aPDT represents a promising novel strategy to improve the clinical treatments of antibiotic-resistant *H. pylori* infection.

## ACKNOWLEDGMENTS

We thank Shanghai Fudan-Zhangjiang Bio-Pharmaceutical Co., Ltd. for providing the LED light systems used *in vivo* and *in vitro*. Furthermore, we thank State Key Laboratory of Microbial Metabolism equipment platform for assistance in flow cytometry and microscopy.

## AUTHOR AFFILIATIONS

[1]School of Life Sciences and Biotechnology, Shanghai Jiao Tong University, Shanghai, People's Republic of China
[2]School of Life Sciences, Fudan University, Shanghai, People's Republic of China

## AUTHOR ORCIDs

Daotan Zhao ⓘ http://orcid.org/0009-0003-9241-0022
Yina Ba ⓘ http://orcid.org/0009-0003-6854-6069
Shuyang Sun ⓘ http://orcid.org/0000-0001-6013-0614

## AUTHOR CONTRIBUTIONS

Daotan Zhao, Data curation, Formal analysis, Methodology, Visualization, Writing – original draft | Yina Ba, Data curation, Formal analysis, Methodology, Visualization, Writing – original draft | Ping Xu, Supervision, Writing – review and editing | Shuyang Sun, Funding acquisition, Methodology, Supervision, Writing – review and editing

## ETHICS APPROVAL

After review by the Animal Ethics Committee of Suzhou Hualian Meide Medical Technology Co., Ltd. on 9 January 2023, it is believed that: This experiment, The effect of photodynamic therapy in treating *Helicobacter pylori* infection in a suckling pig model, complies with the "Regulations on the Management of Experimental Animals" and

the "3R principles." The approval number of the Laboratory Animal Ethics Committee: 230109.

## ADDITIONAL FILES

The following material is available online.

### Open Peer Review

**PEER REVIEW HISTORY (review-history.pdf).** An accounting of the reviewer comments and feedback.

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
