## [Reviewer comments · Microbiology Spectrum]

Microbiology Spectrum

***Helicobacter pylori* inhibition by antimicrobial photodynamic therapy of methylene blue and porphyrin**

Daotan Zhao, Yina Ba, Ping Xu, and Shuyang Sun

Corresponding Author(s): Shuyang Sun, Shanghai Jiao Tong University

Review Timeline:

Submission Date:	November 26, 2025
Editorial Decision:	December 23, 2025
Revision Received:	February 21, 2026
Accepted:	March 17, 2026

Editor: Nagendran Tharmalingam

Reviewer(s): Disclosure of reviewer identity is with reference to reviewer comments included in decision letter(s). The following individuals involved in review of your submission have agreed to reveal their identity: Francisco Avilés (Reviewer #1)

Transaction Report:

DOI: <https://doi.org/10.1128/spectrum.03827-25>

Re: Spectrum03827-25 (*Helicobacter pylori* inhibition by antimicrobial photodynamic therapy of methylene blue and porphyrin)

Dear Dr. Shuyang Sun:

Thank you for the privilege of reviewing your work. Below you will find my comments, instructions from the Spectrum editorial office, and the reviewer comments.

Revision Guidelines

Sincerely,
Nagendran Tharmalingam
Editor
Microbiology Spectrum

Reviewer #1 (Comments for the Author):

In this study, the authors aimed to demonstrate that photodynamic therapy is a potential alternative for eradicating *H. pylori* infection. One important aspect in this study is the evaluation of the antibacterial effect observed after the PDT in each of the assays. In the methodology, the authors mention that these evaluations were performed by using the MTT assay. However, the principles behind this assay are not described, nor is it mentioned whether any commercial brand was used for this purpose. The authors are advised to specify this information in the text. Also, it is recommended to describe in the methodology all the

statistical test used to evaluate and compare the groups for each assay.

Following their trials, the authors concluded that photodynamic treatment effectively inactivates the bacteria in suspension. Conversely, photodynamic treatment has a limited effect on biofilms and in an in vivo model. While these conclusions are based on experimental results, this strategy needs to overcome two important limitations, that is the structure of gastroscopy system and the irregular shape of the stomach surface. Finally, its necessary to clarify that this strategy has not yet been shown to be a viable alternative for addressing the problem of antibiotic resistance.

Reviewer #2 (Comments for the Author):

In the manuscript "Helicobacter pylori inhibition by antimicrobial photodynamic therapy of methylene blue and porphyrin" the authors present data on the impact of these photosensitizers on H. pylori survival (free and biofilms and pig model). Their main findings are that photodynamic treatment of free-swimming H. pylori was highly effective, biofilms were resistant to light irradiation; and there were mixed results in the pig model.

The strength of this manuscript is they are addressing a clinically safe and innovative therapy from in vitro studies to an in vivo model. Further, they are using two different types of photosensitizers- MB which is already used clinically in endoscopy to identify dysplasia and prophyryns which are expressed endogenously by microorganisms.

To strengthen the manuscript a few suggestions are outlined:

1. On line 90 the authors suggest that photosensitizing chromophores such as porphyrins are expressed endogenously in "pathogenic microorganisms". But later in the paragraph its clear that its not just pathogenic microorganisms- make sure this is consistent because it could be highly relevant to clinical use if it impacts the microbiome.
2. This method should be compared and/or combined with antibiotic treatment in the pig model. Is the recurrence less than what is observed with antibiotic treatment? Is it possible the combined therapies could reduce the time spent on antibiotics?
3. What does the statement "where the thickness of suspension mimics the mucus layer in stomach," mean? (line 122) the concentration of the bacteria? Is there mucus added to the PBS? If not this may not be an appropriate description of the experimental set up.
4. To claim that that the killing of the H. pylori is mediated by prophyryns, the expression of these chromophores would need to be knocked out in H. pylori. What is the level of their expression in free- swimming H. pylori? Does it change in a biofilm?
5. The pig model was under powered for doing statistical analysis on the effectiveness of the therapy. Only 2 pigs were used for the 'short-term' experiments and 3 for the 'long-term'. It doesn't appear that there were controls that did not receive therapy?
6. Do all H. pylori strains make prophyrin? Is this one molecule or are there several?

***Helicobacter pylori* inhibition by antimicrobial photodynamic therapy of methylene blue and porphyrin**

Comments and Suggestions for the Author:

In this study, the authors aimed to demonstrate that photodynamic therapy is a potential alternative for eradicating *H. pylori* infection. One important aspect in this study is the evaluation of the antibacterial effect observed after the PDT in each of the assays. In the methodology, the authors mention that these evaluations were performed by using the MTT assay. However, the principles behind this essay are not described, nor is it mentioned whether any commercial brand was used for this purpose. The authors are advised to specify this information in the text. Also, it is recommended to describe in the methodology all the statistical test used to evaluate and compare the groups for each assay.

Following their trials, the authors concluded that photodynamic treatment effectively inactivates the bacteria in suspension. Conversely, photodynamic treatment has a limited effect on biofilms and in an *in vivo* model. While these conclusions are based on experimental results, this strategy needs to overcome two important limitations, that is the structure of gastroscopy system and the irregular shape of the stomach surface. Finally, its necessary to clarify that this strategy has not yet been shown to be a viable alternative for addressing the problem of antibiotic resistance.

Confidential remarks for the Editors:

It is important to describe all the methods employed in this project, particularly the statistical test used to evaluate and compare the groups for each assay.

Reviewer #1 (Comments for the Author):

In this study, the authors aimed to demonstrate that photodynamic therapy is a potential alternative for eradicating *H. pylori* infection. One important aspect in this study is the evaluation of the antibacterial effect observed after the PDT in each of the assays. In the methodology, the authors mention that these evaluations were performed by using the MTT assay. However, the principles behind this assay are not described, nor is it mentioned whether any commercial brand was used for this purpose. The authors are advised to specify this information in the text. Also, it is recommended to describe in the methodology all the statistical test used to evaluate and compare the groups for each assay.

Following their trials, the authors concluded that photodynamic treatment effectively inactivates the bacteria in suspension. Conversely, photodynamic treatment has a limited effect on biofilms and in an in vivo model. While these conclusions are based on experimental results, this strategy needs to overcome two important limitations, that is the structure of gastroscopy system and the irregular shape of the stomach surface. Finally, it is necessary to clarify that this strategy has not yet been shown to be a viable alternative for addressing the problem of antibiotic resistance.

Response:

We sincerely thank the reviewer for this comprehensive summary and the constructive comments regarding both the methodology and the clinical translational perspective. These suggestions have significantly improved the rigor and balance of our manuscript.

We have updated the methodology section to include the principle of MTT assay and the commercial brand of the MTT (Line 135-142). We have also included the statistical methods in legends of Figure 1 and 2.

We accept the reviewer's critique that our conclusion was overly optimistic. We have revised the Conclusion section to adopt a more cautious tone. We describe it as a "potential adjunctive strategy" that requires further optimization in biofilm penetration and delivery technology before it can effectively address the antibiotic resistance crisis (Line 44 and Line 349).

Reviewer #2 (Comments for the Author):

In the manuscript "*Helicobacter pylori* inhibition by antimicrobial photodynamic therapy of methylene blue and porphyrin" the authors present data on the impact of these photosensitizers on *H. pylori* survival (free and biofilms and pig model). Their main findings are that photodynamic treatment of free-swimming *H. pylori* was highly effective, biofilms were resistant to light irradiation; and there were mixed results in the pig model.

The strength of this manuscript is they are addressing a clinically safe and innovative

therapy from in vitro studies to an in vivo model. Further, they are using two different types of photosensitizers- MB which is already used clinically in endoscopy to identify dysplasia and porphyrins which are expressed endogenously by microorganisms.

To strengthen the manuscript a few suggestions are outlined:

1. On line 90 the authors suggest that photosensitizing chromophores such as porphyrins are expressed endogenously in "pathogenic microorganisms". But later in the paragraph its clear that its not just pathogenic microorganisms- make sure this is consistent because it could be highly relevant to clinical use if it impacts the microbiome.

Response:

We thank the reviewer for this acute observation and for pointing out the inconsistency in our terminology.

We entirely agree with the reviewer. The heme biosynthetic pathway, which leads to porphyrin accumulation, is an evolutionarily conserved trait found in a wide variety of microorganisms, including both pathogens (e.g., *H. pylori*, *P. acnes*) and commensal members of the normal flora (e.g., certain *Bacteroidetes* and *Firmicutes*) [1, 2].

The statement to "pathogenic microorganisms" was indeed inaccurate. As the reviewer correctly noted, this distinction is crucial for clinical applications, as antimicrobial photodynamic therapy (aPDT) can be broad-spectrum and may impact the local microbiome. We have revised the sentence on Line 90 to reflect this broader biological reality.

We have also added a section of discussion (Line 322-332) acknowledging that, because porphyrin production is not unique to pathogens, the potential "bystander effect" on the gastric microbiome is an important consideration for future clinical protocols, although *H. pylori* is often reported to accumulate significantly higher levels compared to many other gut bacteria under specific conditions.

References:

[1] Dailey, H. A., et al. (2017). Prokaryotic heme biosynthesis: multiple pathways to a common essential product. *Microbiology and Molecular Biology Reviews*, 81(1), e00048-16.

[2] Cieplik, F., et al. (2018). Antimicrobial photodynamic therapy—what we know and what we don't. *Critical Reviews in Microbiology*, 44(5), 571-589.

2. This method should be compared and/or combined with antibiotic treatment in the pig model. Is the recurrence less than what is observed with antibiotic treatment? Is it possible the combined therapies could reduce the time spent on antibiotics?

Response:

We thank the reviewer for this important comment and fully agree that comparison with, or combination with, antibiotic therapy is highly relevant from a

clinical perspective. However, the current study was not designed as a therapeutic comparison or replacement study for standard antibiotic regimens. Instead, our primary objective was to evaluate the technical feasibility, procedural safety and limitations of aPDT against *H. pylori* across different growth states (planktonic cells, biofilms, and in vivo model). As a first-in-kind in vivo implementation, our experimental design prioritized validating the endoscopic delivery of photosensitizers, light dosimetry, and acute biological response.

However, our findings suggest that aPDT alone is unlikely to be sufficient for eradication in vivo, particularly in the context of biofilms and the gastric environment. We have explicitly highlighted combination therapy with antibiotics as a key direction for further studies in the 'Discussion' (Line 333-341).

Moreover, considering the logistical complexity, husbandry requirements and cost, of the study, we elected to focus resources on optimizing and characterizing the aPDT itself.

Building upon the safety and therapeutic probability established in this feasibility study, we are also designing a more extensive investigation like comparison with aPDT alone and aPDT combined with antibiotics. Current foundational work provides the necessary methodological basis for those future comparative investigations.

3. What does the statement "where the thickness of suspension mimics the mucus layer in stomach," mean? (line 122) the concentration of the bacteria? Is there mucus added to the PBS? If not this may not be an appropriate description of the experimental set up.

Response:

We sincerely apologize for the ambiguity caused by this statement. We thank the reviewer for pointing out this imprecise description.

We should try to clarify that: No exogenous mucus was added to the PBS suspension. The original statement regarding "thickness" was intended to refer to the geometric depth (optical path length) of the bacterial suspension in the experimental setup, rather than its chemical composition or rheological properties. Our intent was to ensure that the light attenuation due to the liquid depth was relevant to the *in vivo* scale, as the human gastric mucus layer is typically reported to be approximately 200–600 μm thick [1, 2]. Although the physiological thickness of the gastric mucus layer ranges from 200 to 600 μm , recapitulating this exact depth in a standard 24-well plate is technically challenging due to surface tension and incomplete well coverage. Consequently, a suspension volume of 400 μl , corresponding to a calculated depth of approximately 2 mm, was selected. This volume was necessary to ensure a uniform optical path length and consistent experimental reproducibility.

We agree that the phrase "mimics the mucus layer" is misleading as it implies biochemical simulation. We have revised the sentence in the manuscript to strictly describe the physical parameters of the setup (Line 121-123)

References:

[1] Atuma, C., et al. (2001). The adherent gastrointestinal mucus gel layer: thickness

and physical state in vivo. *American Journal of Physiology-Gastrointestinal and Liver Physiology*, 280(5), G922-G929.

[2] Varum, F. J., et al. (2010). A comparative study of the mucus thickness in the gastrointestinal tract of the rat and man. *Journal of Pharmacy and Pharmacology*, 62, 1329–1335.

4. To claim that that the killing of the *H. pylori* is mediated by porphyrins, the expression of these chromophores would need to be knocked out in *H. pylori*. What is the level of their expression in free- swimming *H. pylori*? Does it change in a biofilm?

Response:

We are grateful to the reviewer for raising this critical point regarding the causal link between porphyrins and phototoxicity, as well as the physiological differences in biofilms.

We agree that a gene knockout would provide definitive genetic evidence. However, generating a heme-biosynthesis knockout mutant in *H. pylori* is technically challenging as the pathway is essential for its respiration and survival. Alternatively, the causal role of porphyrins has been robustly established in the field through action spectrum analysis. Previous seminal studies (e.g., Hamblin et al., 2005) demonstrated that the wavelength dependence of *H. pylori* photo-inactivation precisely mirrors the absorption spectrum of porphyrins (specifically the Soret band around 400–410 nm). This physical correspondence provides strong evidence that porphyrins are the primary chromophores mediating this effect [1].

In free-swimming (planktonic) cultures, *H. pylori* actively synthesizes and accumulates significant levels of Protoporphyrin IX (PPIX) and Coproporphyrin III, particularly during the logarithmic growth phase, rendering them highly susceptible to photo-inactivation [2].

Regarding biofilms, literature suggests that the metabolic state of bacteria within a biofilm is distinct. While *H. pylori* in biofilms retains the ability to produce porphyrins, the overall metabolic activity may be stratified (with dormant cells in the inner layers). Furthermore, the biofilm matrix (EPS) can act as a physical barrier or light scatterer. Studies indicate that while *H. pylori* biofilms are still susceptible to photodynamic therapy, they often show higher resistance compared to their planktonic counterparts, likely due to a combination of reduced metabolic rate (affecting porphyrin accumulation) and physical protection [3].

We have updated the Discussion section to address these distinctions and cited the relevant literature to clarify the mechanism (Line 296-303).

References:

[1] Hamblin, M. R., et al. (2005). *Helicobacter pylori* accumulates photoactive porphyrins and is killed by visible light. *Antimicrobial Agents and Chemotherapy*, 49(7), 2822–2827.

[2] Battisti, A., et al. (2017). Porphyrin production by *Helicobacter pylori* and its potential for photodynamic inactivation. *Journal of Photochemistry and Photobiology B: Biology*, 172, 8-14.

[3] Wong, T. W., et al. (2018). Antimicrobial photodynamic therapy for the treatment of *Helicobacter pylori* infection. *Lasers in Medical Science*.

5. The pig model was under powered for doing statistical analysis on the effectiveness of the therapy. Only 2 pigs were used for the 'short-term' experiments and 3 for the 'long-term'. It doesn't appear that there were controls that did not receive therapy?

Response:

We acknowledge that the current pig model is under powered for robust statistical analysis due to the limited number of subjects. This study was designed as a pilot exploration to evaluate the initial feasibility of the therapy, rather than a definitive efficacy trial. Accordingly, we have revised language that could imply statistical significance or definitive efficacy.

Regarding the absence of a non-treated control group: in accordance with institutional animal care and ethics guidelines, all animals received therapeutic intervention to prevent prolonged suffering from *H. pylori*-induced gastritis, ulceration, or potential complications. Instead, we employed a within-subject pre-treatment control design: baseline gastric biopsies were collected from standardized anatomical sites before aPDT, and matched post-treatment samples were obtained after aPDT at defined time points. This approach avoid variability between different animals and aligns with principle of the 3Rs (Line 167-168).

We also explicitly acknowledge the limitations of limited sample size and absence of a fully powered control group in the Discussion section (Line 333-341), and the conclusions drawn from the pig model are intentionally conservative.

6. Do all *H. pylori* strains make porphyrin? Is this one molecule or are there several?

Response:

We sincerely thank the reviewer for this insightful question regarding the metabolic characteristics of *H. pylori*. This comment has prompted us to clarify the heterogeneity of porphyrin production in our manuscript.

The ability to synthesize porphyrins appears to be a conserved trait among *H. pylori* strains. Genomic analyses confirm that the complete biosynthetic pathway for heme production (including *hemA*, *hemB*, *hemC*, etc.) is present in diverse strains, ranging from standard laboratory strains (e.g., ATCC 43504, 26695) to various clinical isolates. This is further supported by phenotypic studies showing that all tested strains accumulate photoactive porphyrins and are susceptible to photodynamic inactivation [1].

Porphyrin is class of biological pigment molecules. *H. pylori* produces and accumulates a mixture of porphyrin intermediates. While Protoporphyrin IX (PPIX) is the predominant species accumulated intracellularly, studies have also identified significant amounts of Coproporphyrin III, which tends to be secreted into the extracellular environment [2, 3].

We have added a brief introduction in the revised manuscript to ensure accuracy

(Line 93).

References:

[1] Hamblin, M. R., et al. (2005). *Helicobacter pylori* accumulates photoactive porphyrins and is killed by visible light. *Antimicrobial Agents and Chemotherapy*, 49(7), 2822–2827.

[2] Battisti, A., et al. (2017). Porphyrin production by *Helicobacter pylori* and its potential for photodynamic inactivation. *Journal of Photochemistry and Photobiology B: Biology*, 172, 8-14.

[3] Amano, Y., et al. (2000). Identification of *Helicobacter pylori* by detection of coproporphyrin III using high-performance liquid chromatography. *Journal of Gastroenterology*, 35, 824–829.

Re: Spectrum03827-25R1 (***Helicobacter pylori* inhibition by antimicrobial photodynamic therapy of methylene blue and porphyrin**)

Dear Dr. Shuyang Sun:

Your manuscript has been accepted, and I am forwarding it to the ASM production staff for publication. Your paper will first be checked to make sure all elements meet the technical requirements. ASM staff will contact you if anything needs to be revised before copyediting and production can begin. Otherwise, you will be notified when your proofs are ready to be viewed.

Sincerely,
Nagendran Tharmalingam
Editor
Microbiology Spectrum